# Strontium Binding to α-Parvalbumin, a Canonical Calcium-Binding Protein of the “EF-Hand” Family

**DOI:** 10.3390/biom11081158

**Published:** 2021-08-05

**Authors:** Alisa A. Vologzhannikova, Marina P. Shevelyova, Alexey S. Kazakov, Andrey S. Sokolov, Nadezhda I. Borisova, Eugene A. Permyakov, Nikoleta Kircheva, Valya Nikolova, Todor Dudev, Sergei E. Permyakov

**Affiliations:** 1Institute for Biological Instrumentation, Pushchino Scientific Center for Biological Research of the Russian Academy of Sciences, Pushchino, 142290 Moscow, Russia; lisiks.av@gmail.com (A.A.V.); marina.shevelyova@gmail.com (M.P.S.); fenixfly@yandex.ru (A.S.K.); 212sok@gmail.com (A.S.S.); niborisova@list.ru (N.I.B.); epermyak@yandex.ru (E.A.P.); 2Institute of Optical Materials and Technologies “Acad. J. Malinowski”, Bulgarian Academy of Sciences, 1113 Sofia, Bulgaria; nkircheva@iomt.bas.bg; 3Faculty of Chemistry and Pharmacy, Sofia University “St. Kl. Ohridski”, 1164 Sofia, Bulgaria; ohtvd@chem.uni-sofia.bg (V.N.); t.dudev@chem.uni-sofia.bg (T.D.)

**Keywords:** metal selectivity, calcium-binding proteins, EF-hand, parvalbumin, protein structure, protein stability

## Abstract

Strontium salts are used for treatment of osteoporosis and bone cancer, but their impact on calcium-mediated physiological processes remains obscure. To explore Sr^2+^ interference with Ca^2+^ binding to proteins of the EF-hand family, we studied Sr^2+^/Ca^2+^ interaction with a canonical EF-hand protein, α-parvalbumin (α-PA). Evaluation of the equilibrium metal association constants for the active Ca^2+^ binding sites of recombinant human α-PA (‘CD’ and ‘EF’ sites) from fluorimetric titration experiments and isothermal titration calorimetry data gave 4 × 10^9^ M^−1^ and 4 × 10^9^ M^−1^ for Ca^2+^, and 2 × 10^7^ M^−1^ and 2 × 10^6^ M^−1^ for Sr^2+^. Inactivation of the EF site by homologous substitution of the Ca^2+^-coordinating Glu in position 12 of the EF-loop by Gln decreased Ca^2+^/Sr^2+^ affinity of the protein by an order of magnitude, whereas the analogous inactivation of the CD site induced much deeper suppression of the Ca^2+^/Sr^2+^ affinity. These results suggest that Sr^2+^ and Ca^2+^ bind to CD/EF sites of α-PA and the Ca^2+^/Sr^2+^ binding are sequential processes with the CD site being occupied first. Spectrofluorimetric Sr^2+^ titration of the Ca^2+^-loaded α-PA revealed presence of secondary Sr^2+^ binding site(s) with an apparent equilibrium association constant of 4 × 10^5^ M^−1^. Fourier-transform infrared spectroscopy data evidence that Ca^2+^/Sr^2+^-loaded forms of α-PA exhibit similar states of their COO^−^ groups. Near-UV circular dichroism (CD) data show that Ca^2+^/Sr^2+^ binding to α-PA induce similar changes in symmetry of microenvironment of its Phe residues. Far-UV CD experiments reveal that Ca^2+^/Sr^2+^ binding are accompanied by nearly identical changes in secondary structure of α-PA. Meanwhile, scanning calorimetry measurements show markedly lower Sr^2+^-induced increase in stability of tertiary structure of α-PA, compared to the Ca^2+^-induced effect. Theoretical modeling using Density Functional Theory computations with Polarizable Continuum Model calculations confirms that Ca^2+^-binding sites of α-PA are well protected against exchange of Ca^2+^ for Sr^2+^ regardless of coordination number of Sr^2+^, solvent exposure or rigidity of sites. The latter appears to be a key determinant of the Ca^2+^/Sr^2+^ selectivity. Overall, despite lowered affinity of α-PA to Sr^2+^, the latter competes with Ca^2+^ for the same EF-hands and induces similar structural rearrangements. The presence of a secondary Sr^2+^ binding site(s) could be a factor contributing to Sr^2+^ impact on the functional activity of proteins.

## 1. Introduction

Strontium, a non-biogenic element from group II of the periodic table, positioned under its biogenic congener, calcium, is clinically used as a therapeutic agent. It prevents the destruction of bones and contributes to their restoration [1,2,3,4,5,6,7]. Sr^2+^ salt of ranelic acid is a medication (under the names Protelos, Protos, Strontium ranelate Aristo) used for treatment of osteoporosis in postmenopausal women and very elderly patients. However, strontium ranelate has serious side effects: cardiovascular disorders (including myocardial infarction), venous thromboembolism, pulmonary embolism, nausea, diarrhea, headache and eczema. It is prescribed only as a second-line drug for treatment of postmenopausal osteoporosis when other drugs are unsuitable [4,8]. Furthermore, ^89^Sr chloride (half-life of 50.6 days) has been efficient in treatment of bone cancer patients [9], since strontium as a calcium mimetic accumulates at sites of enhanced osteogenesis [6], thereby exposing cancerous formations to radiation.

Strontium’s similarity to calcium is critical for its physiological action. Their ionic radii (for heptacoordinated metal) are 1.06 Å for Ca^2+^ and 1.21 Å for Sr^2+^ [10]. Ca^2+^ and Sr^2+^ are ‘hard’ cations with strong preference toward oxygen-containing ‘hard’ ligands. Both cations are characterized by flexible coordination/hydration spheres comprising six to nine ligands [11,12]. 

Although much information has been accumulated on the beneficial effects of Sr^2+^ on human health, the intimate mechanisms of its therapeutic action remain poorly understood. The widely accepted explanation of its curative effects postulates that Sr^2+^ is able to bind and activate the calcium-sensing receptor (CaSR), a representative of G protein-coupled receptors [1,2,3,4,6,8,13,14]. CaSR activation triggers a cascade of signaling events promoting both osteoclast apoptosis and osteoblast differentiation. Meanwhile, the abovementioned side effects suggest that Sr^2+^ has other targets in human body, likely related to calcium homeostasis. Therefore, it is of particular importance to study interactions of Sr^2+^ with Ca^2+^-binding proteins, which could be attacked by Sr^2+^, to elucidate the competition between the cations. This knowledge is of importance for exploring the pathways of ^90^Sr propagation in human body (with a half-life 28.8 years), as it is a substantial component of radioactive waste from nuclear reactors.

The largest family of eukaryotic Ca^2+^-binding proteins is the EF-hand superfamily [15]. EF-hand proteins contain typically paired EF-hand motifs, which represent a 12-residue Ca^2+^-coordinating loop flanked by α-helices (‘helix-loop-helix’ motif, see the [1] PDOC00018 entry of PROSITE). Among the hundreds of known EF-hand proteins, parvalbumin (PA) serves a reference point, since it was the first EF-hand protein whose structure was solved [16]. PA is a 10–12 kDa globular α-helical protein of vertebrates, consisting of three EF-hand domains, two of which (CD and EF) bind Ca^2+^/Mg^2+^ with the highest affinity among EF-hand proteins (reviewed in [17,18,19]). Equilibrium Ca^2+^ association constants of PA measured at room temperatures lie in the range of 10^8^ to 10^10^ M^−1^, while the respective equilibrium Mg^2+^ binding constants are within the 10^3^ to 10^5^ M^−1^ range [19]. Several studies suggest a sequential mechanism of Ca^2+^/Mg^2+^ binding to PA [18,19]. The CD site seems to be filled by Ca^2+^ first, followed by filling of the EF site. Meanwhile, the data on Sr^2+^ affinity of PA and the mechanism of Sr^2+^ binding have not been reported to date, to the best of our knowledge. Therefore, in the present work we focus on the Sr^2+^/Ca^2+^-binding properties of recombinant human α-PA, and seek to address the following intriguing questions: (1) To what extent are the Ca^2+^-binding sites of PA prone to Sr^2+^ binding? (2) What are the key factors governing the competition between Ca^2+^/Sr^2+^ in PA? (3) Do secondary Sr^2+^-binding sites, differing from CD and EF sites, exist in PA? The results obtained help elucidate the intimate mechanisms of metal selectivity in PA and to reveal its thermodynamic determinants.

## 2. Materials and Methods

### 2.1. Materials

Ultra-grade HEPES (N-(2-hydroxyethyl)piperazine-N′-(2-ethanesulfonic acid)) and H_3_BO_3_ were from Sigma-Aldrich Co. (St. Louis, CA, USA) and Fluka (Munich, Germany), respectively. Biochemical grade Tris (tris(hydroxymethyl)aminomethane) and ultra-grade TCA (trichloroacetic acid) were purchased from Panreac AppliChem (Darmstadt, Germany). Calcium chloride standard solution was from Fluka (Munich, Germany). Strontium chloride standard solution, ultra-grade KOH and EDTA (ethylenediaminetetraacetic acid) acid were from Sigma–Aldrich Co. (St. Louis, CA, USA). Standard solutions of EDTA potassium salt was prepared as described in [20]. TOYOPEARL SuperQ-650M was purchased from Tosoh Bioscience (Griesheim, Germany). HiPrep 26/60 Sephacryl S-100 HR was from GE Healthcare (Chicago, IL, USA). Sephadex G-25 was a product of Pharmacia LKB (Uppsala, Sweden). A molecular mass marker for SDS-PAGE was a product of Thermo Scientific (Waltham, MA, USA). d-10-camphorsulfonic acid was from JASCO, Inc. (Tokyo, Japan). Other chemicals were reagent grade or higher. All buffers and other solutions were prepared using ultrapure water (Millipore Simplicity 185 system). Plastics or quartz ware were used instead of glassware, to avoid contamination of protein samples with Ca^2+^. SnakeSkin dialysis tubing (3.5 kDa MWCO) from Thermo Scientific (Waltham, MA, USA) was used.

### 2.2. Expression and Purification of Human α-PA

Human α-PA gene (Swiss-Prot entry P20472) was obtained from DNASU Plasmid Repository (https://dnasu.org/DNASU/, (accessed on 14 July 2021)), clone HsCD00301153, and cloned into the pET11a vector between the NdeI and BamHI restriction sites. The plasmid was transformed into *E. coli* strain BL21 (DE3). rWT α-PA (recombinant wild-type α isoform parvalbumin) was expressed and purified according to [21] with the following modifications. The cells were lysed using a French press (IBI RAS, Pushchino, Russia) and centrifuged at 30,000× *g* for 30 min at 4 °C; 1 mM CaCl_2_ was added to the supernatant and the solution was kept at 80 °C for 3 min, cooled using ice-water bath and centrifuged at 15,500× *g* for 30 min at 4 °C. The supernatant was dialyzed against distilled water at 4 °C, followed by dialysis against 10 mM Tris-HCl, pH 8.5 buffer. The dialyzed solution was passed through TOYOPEARL SuperQ-650M column (0.9 cm × 7.5 cm) equilibrated with the same buffer. PA was eluted with a linear gradient of NaCl (75 mL of 0 mM NaCl to 75 mL of 100 mM NaCl); the elution rate was 1 mL/min. The fractions containing PA were joined, precipitated by 3% TCA, incubated at 4 °C for 10 min and centrifuged at 5000× *g* for 10 min at 4 °C. The pellet was resuspended in 50 mM Tris-HCl pH 8.0 buffer and passed through HiPrep 26/60 Sephacryl S-100 HR gel filtration column equilibrated with 10 mM Tris-HCl pH 7.5 buffer (flow rate of 1 mL/min). The fractions with PA were collected, exhaustively dialyzed against distilled water at 4 °C, freeze-dried and stored at −18 °C. The yield was 40–60 mg of PA per liter of cell culture.

Site-directed mutagenesis of human α-PA was carried out according to [22]. The resulting PA mutants with inactivated EF-loops, PA_-CD_ and PA_-EF_, caused by the substitutions E63Q and E102Q, respectively, were expressed and purified similarly to rWT α-PA.

The purified PA samples were homogeneous as judged from native and SDS-PAGE (sodium dodecyl sulfate polyacrylamide gel electrophoresis). Their UV absorption and fluorescence spectra were characteristic for Phe residues, since human α-PA lacks Trp/Tyr residues. PA concentrations were estimated spectrophotometrically using the ε_259nm_ values derived from the molar extinction coefficient at 205 nm, and were estimated according to [23] as 1585 M^−1^cm^−1^ for rWT α-PA, 1672 M^−1^cm^−1^ for PA_-CD_ and 1359 M^−1^cm^−1^ for PA_-EF,_ respectively.

### 2.3. Removal of Metal Ions from PA Samples

The contaminating Ca^2+^/Mg^2+^ ions were removed from PA by precipitation of the protein by TCA [24], followed by gel-filtration according to [25] using Sephadex G-25 column equilibrated with a respective buffer.

### 2.4. Fluorescence Measurements

Fluorescence emission spectra were measured using a Cary Eclipse spectrofluorimeter (Varian, Inc., Palo Alto, CA, USA) equipped with a Peltier-controlled cell holder. Quartz cells with pathlengths of 10 mm were used. Protein concentrations were 1–19 µM. Buffer conditions: 10 mM HEPES-KOH, pH 8.1. Fluorescence of Phe residues of PA was excited at 259 nm.

### 2.5. Fluorimetric Evaluation of Ca^2+^/Sr^2+^ Affinity of PA

The metal affinity of PA at 20 °C was estimated from spectrofluorimetric titration of the metal-depleted protein (1–19 µM) with a Ca^2+^/Sr^2+^ standard stock solution, followed in some cases by a titration of the metal-loaded protein with EDTA standard solution. Buffer conditions: 10 mM HEPES–KOH, pH 8.1. Fluorescence intensities were corrected for the dilution effect via division of the experimental values by a factor of (1–10*^−Dexc^*), where *D_exc_* is the protein absorption at the excitation wavelength.

The experimental data for Ca^2+^ binding to rWT α-PA were approximated by the sequential metal binding scheme, previously shown to be valid for various PAs [19,26,27,28]:*K_Me_*_1_Protein + Me^2+^ ↔ Protein•Me^2+^*K_Me_*_2_Protein•Me^2+^ + Me^2+^ ↔ Me^2+^•Protein•Me^2+^,(1)
where Me^2+^ denotes the metal ion; *K_Me_*_1_ and *K_Me_*_2_ are equilibrium metal association constants for the two active EF-hands of the protein.

Upon use of EDTA, the competition between protein and EDTA for metal ions was taken into account:*K_EDTA_* EDTA + Me^2+^ ↔ EDTA•Me^2+^(2)

The Ca^2+^/Sr^2+^ association constants of EDTA, *K_EDTA_*, were calculated according to [29].

In the case of Ca^2+^ binding to rWT α-PA, the data were globally fitted according to the schemes (1) and (2) using FluoTitr v.1.42 software (IBI RAS, Pushchino, Russia), implementing the nonlinear regression algorithm by Marquardt [30]. *K_Me_*_1_, *K_Me_*_2_ and specific fluorescence intensities of the possible protein states were used as fitting parameters. The accuracy of the *K_Me_*_1_ and *K_Me_*_2_ estimates was about ±1/4 orders of their magnitudes.

In some cases the same approach was applied, but using the single site metal binding scheme:*K_Me_* Protein + Me^2+^ ↔ Protein•Me^2+^(3)
where Me^2+^ denotes metal ion, *K_Me_* is an equilibrium metal association constant.

### 2.6. ITC Measurements

ITC (isothermal titration calorimetry) studies were carried out at 20 °C using Nano ITC calorimeter (TA Instruments, New Castle, DE, USA). The reference cell was filled with deionized water. The reaction cell contained 1000 μL of 97 μM PA in 10 mM HEPES-KOH pH 8.1 buffer. The PA solution was titrated by 2.52 μL aliquots of 3.6 mM SrCl_2_ in the same buffer: 37 injections with a 200 s interval between the injections. The injection syringe was rotated at 280 rpm during the titration and pre-equilibration. The ITC data were fitted by the sequential metal binding scheme (1) using FluoTitr v.1.42 software (IBI RAS, Pushchino, Russia), implementing the nonlinear regression algorithm by Marquardt [30]. *K_Me_*_1_, *K_Me_*_2_ and specific enthalpies of the possible protein states were used as fitting parameters.

### 2.7. FTIR Spectroscopy Measurements

Transmission FTIR spectroscopy (Fourier-transform infrared spectroscopy) spectra of PA (3 mM) were measured at 20 °C using Nicolet 6700 spectrometer (Thermo Scientific), equipped with a MCT detector cooled by liquid N_2_. The measurements were performed in a CaF_2_ cell with a pathlength of 4 µm in 10 mM H_3_BO_3_-KOH pH 9.0 buffer, either without metal ions or in the presence of 10 mM CaCl_2_/SrCl_2_. Apo-α-PA (metal-free form of α-PA) was prepared using combination of the TCA protein precipitation method [24] and Sephadex G-25 gel-filtration method [25]. Resolution was 2 cm^−1^; 256 spectra were averaged and the averaged spectrum of the buffer was subtracted. The difference spectrum was smoothed by the FFT algorithm with a window of 20 points and 0.02593 cutoff frequency; its second derivative was calculated using OriginPro v.9.0 software (OriginLab Corp., Northampton, MA, USA).

### 2.8. Circular Dichroism Measurements

CD spectra of PA were measured at 20 °C with a J-810 spectropolarimeter (JASCO, Inc., Tokyo, Japan) equipped with a Peltier-controlled cell holder. The instrument was calibrated with an aqueous solution of d-10-camphorsulfonic acid according to the manufacturer’s instruction. The cell compartment was purged with N_2_. Cells with pathlengths of 10 mm and 1.00 mm were used for near- and far-UV regions, respectively. Protein concentrations were 6.2–7.5 and 126–129 µM for far- and near-UV regions, respectively. Buffer conditions: 10 mM H_3_BO_3_-KOH, pH 9.0, 1 mM EDTA or 1 mM SrCl_2_/CaCl_2_. Bandwidth was 2 nm, with an average time of 1–2 s and an accumulation of 3. Buffer contribution was subtracted from PA spectra. Estimates of protein secondary structure contents were carried out using the CDPro software package [31]. The experimental data in the 195–240 nm range were treated by SELCON3, CDSSTR and CONTIN algorithms, using SDP48 and SMP56 reference protein sets. The final secondary structure fractions represented averaged values.

### 2.9. Scanning Calorimetry Measurements

DSC (differential scanning calorimetry) studies were carried out using a DASM-4M differential scanning microcalorimeter (IBI RAS, Pushchino, Russia) at a heating rate of 0.86 K/min and an excess pressure of 3 bar. PA concentrations were 1.2–1.8 mg/mL. Buffer conditions: 10 mM H_3_BO_3_-KOH, pH 9.0, 1 mM SrCl_2_/CaCl_2_ or without additions. The measurements and calculations of protein specific heat capacity (*C_p_*) were performed as described in [27]. The specific heat capacity of the fully unfolded protein was estimated according to [32]. The temperature dependence of *C_p_* was analyzed according to the cooperative two-state model:*n*∙N ↔ *n*∙D(4)
where N and D denote native and denatured protein states, respectively, and *n* is a number of protein molecules involved into transition. The experimental data were fitted by the equations derived earlier [33] using OriginPro v.9.0 software (OriginLab Corp., Northampton, MA, USA). The heat capacity change accompanying the thermal transition (ΔC_p_) was supposed to be independent of temperature. ΔC_p_, mid-transition temperature (T_0_), enthalpy of protein denaturation at temperature T_0_ (ΔH_0_) and n were used as fitting parameters.

### 2.10. Models Used in Theoretical Calculations

The models of Ca^2+^/Sr^2+^-loaded CD and EF sites of PA were built based on the high-resolution X-ray structure of Ca^2+^-bound pike β-PA (PDB entry 1PAL; resolution 1.65 Å) [34]. The metal ion and its first-shell ligands (Asp51, Asp53, Glu59, Glu62, Ser55 and Phe57^backbone^ for the CD site; Asp90, Asp92, Asp94, Met96^backbone^, Glu101 and Water246 for the EF site) were incised from the protein structure and capped at the C^α^ atoms with a methyl group. Thus, the side chains of Asp^−^, Glu^−^ and Ser were represented by CH_3_CH_2_COO^−^, CH_3_CH_2_CH_2_COO^−^ and CH_3_CH_2_OH, respectively, while the metal-coordinating backbone peptide group was modeled by N-methylacetamide (CH_3_CONHCH_3_). The side chains of Glu62 (CD site) and Glu101 (EF site) bind Ca^2+^ in a bidentate fashion, whereas the rest of carboxylic residues coordinate the metal monodentately. The coordination number of Ca^2+^ in the both sites was seven. The resulting Ca^2+^-bound constructs were subjected to geometry optimization (see below). Ca^2+^ of the optimized structures was replaced by Sr^2+^ and the resulting Sr^2+^-containing constructs were fully optimized. Additionally, the metal-binding sites were modified (see Section 3) and accordingly optimized to access the role of several factors (ligand mutations or increased coordination number of Sr^2+^) in the site metal selectivity.

The competition between Sr^2+^ and Ca^2+^ can be expressed in terms of the change in Gibbs free energy upon replacement of the cognate Ca^2+^ cation bound to the site by Sr^2+^:[Sr^2+^-aq] + [Ca^2+^-protein] → [Sr^2+^-protein] + [Ca^2+^-aq](5)
where [Ca^2+^/Sr^2+^-protein] and [Ca^2+^/Sr^2+^-aq] represent the metal cation bound to protein ligands within the binding pocket and the unbound metal, respectively. A positive free energy change for Equation (5) implies a Ca^2+^-selective site, whereas a negative value suggests a Sr^2+^-selective one.

### 2.11. DFT/PCM Calculations (Density Functional Theory/Polarizable Continuum Model)

The M06-2X method [35] in combination with Pople’s triple zeta 6-311++G(d,p) basis set for C, H, N, O and C^α^ atoms and the SDD basis set/effective core potential for Sr^2+^ was employed in the calculations. This theoretical method/basis set has been thoroughly calibrated/validated in previous publications with respect to available experimental data and has proven to be dependable, since it can properly reproduce the geometry of a series of representative metal structures [36] as well as the free energies of metal substitution in acetate, imidazole and glycine complexes [37]. Furthermore, it was additionally validated here with respect to experimentally determined free energies of Sr^2+^ Ca^2+^ exchange in metal-oxalate complexes (see below). 

Each metal-loaded binding site was optimized in the gas phase by employing the Gaussian 09 software package [38]. Electronic energies, E_el_, were computed for each optimized complex. Subsequent frequency calculations were performed at the same M06-2X/6-311++G(d,p)//SDD level of theory, and no imaginary frequency was found for any of the structures studied. The frequencies were scaled by an empirical factor of 0.983 [39] and employed to calculate the thermal energies, E_th_, including zero-point energy and entropies, S. The differences (i.e., ΔE_el,_ ΔE_th_ and ΔS) between the products and reactants in Equation (5) were used to evaluate the metal exchange Gibbs free energy in the gas phase at T = 298.15 K, ΔG^1^, according to the equation:ΔG^1^ = ΔE_el_^1^ + ΔE_th_^1^ − TΔS^1^(6)

The basis set superposition error for Equation (5) was found to be negligible [40], and therefore was not considered in the present calculations. 

Metal-binding sites are located in cavities/clefts of proteins with dielectric properties that are different from those of the bulk water [41] but similar to those of the low-polarity solvents [42]. Therefore, condensed-phase evaluations were performed in solvents mimicking the dielectric properties of buried and solvent-exposed binding sites, diethyl ether (dielectric constant ε = 4) and propanonitrile (ε = 29), respectively. Solvation effects were taken into consideration by using the solvation model based on density (SMD) scheme [43], as implemented in the Gaussian 09 software. The optimized structure of each metal construct in the gas phase was subjected to single point calculations in the respective solvent at the M06-2X/6-311++G(d,p)//SDD level of theory. The difference between the gas-phase and SMD energies was used to compute the solvation free energy, ΔG_solv_^ε^, of each complex. The incoming Sr^2+^ and outgoing Ca^2+^ ions were considered to be in a bulk aqueous environment (ε = 78) outside the binding pocket. Accordingly, their experimentally evaluated hydration free energies of −329.8 and −359.7 kcal/mol, respectively [44], were employed in the calculations. The free energy of cation exchange in a protein cavity characterized by an effective dielectric constant ε was evaluated as follows:ΔG^ε^ = ΔG^1^ + ΔG_solv_^ε^ ([Sr^2+^-protein]) − ΔG_solv_^ε^ ([Ca^2+^-protein]) − ΔG_solv_^78^ ([Sr^2+^-aq]) + ΔG_solv_^78^ ([Ca^2+^-aq])(7)

### 2.12. Validation of Calculations

The theoretical method and calculation protocol employed here were validated with respect to experimental data on Sr^2+^ → Ca^2+^ substitution in metal complexes. Experimentally measured stability constants of Ca^2+^ and Sr^2+^ oxalate complexes [45] were used to evaluate the free energy of metal exchange in a water environment:[Sr^2+^-aq ]+ [Ca^2+^(H_2_O)_5_(C_2_O_4_^2-^)]^0^ → [Sr^2+^(H_2_O)_5_(C_2_O_4_^2-^)]^0^ + [Ca^2+^-aq](8)

Note that the two metals were bound bidentately to the oxalate anion and characterized by a coordination number of 7. The theoretically evaluated free energy of metal substitution of 1.4 kcal/mol was in very good agreement with the experimental estimate of 0.6 kcal/mol.

## 3. Results

### 3.1. Ca^2+^/Sr^2+^-Binding Properties of Human α-Parvalbumin

The recombinant wild-type human α-parvalbumin (rWT α-PA) was expressed in *E. coli* strain BL21(DE3), isolated and purified to homogeneity mainly according to [21], with some modifications. Since rWT α-PA lacks W/Y residues, but possesses nine Phe residues, we monitored the Ca^2+^-binding process via Phe fluorescence (Figure 1A). Addition of Ca^2+^ to apo-α-PA was accompanied by a decrease in fluorescence intensity of PA up to a Ca^2+^ to PA molar ratio of 2, followed by a plateau. The further addition of EDTA, a strong Ca^2+^ chelator, inverted the effect. Thus, Phe fluorescence of rWT α-PA revealed the binding of two Ca^2+^ ions per protein molecule.

Impairment of the Ca^2+^-binding loop in the EF domain of α-PA (EF site) by the homologous E102Q substitution in 12th position of the loop (PA_-EF_ mutant) caused a decline in the stoichiometry of Ca^2+^ binding to PA down to 1:1, and facilitated Ca^2+^ removal via EDTA (Figure 1B). Meanwhile, the analogous damage of the Ca^2+^-binding loop in the CD domain of α-PA (CD site) via E63Q substitution (PA_-CD_ mutant) induced a drastic decrease in apparent Ca^2+^ affinity of the protein and failure to achieve the fluorescence intensity characteristic of the Ca^2+^-loaded state (Figure 1C). Hence, we may presume the CD site is critical for maintenance of Ca^2+^ affinity of the protein, while the EF site is less significant. Overall, these experiments provide evidence that binding Ca^2+^ ions to α-PA is a sequential process, with the CD site being occupied first. Indeed, the fluorescence titration data for rWT α-PA were successfully described within the sequential metal binding scheme (1), considering a competition between the protein and EDTA for Ca^2+^ (scheme (2)), as shown in Figure 1A and Table 1. The data for PA_-EF_ and PA_-CD_ mutants were approximated by the single site metal binding scheme (3) (Figure 1B,C; Table 1). The inactivation of the EF site suppressed Ca^2+^ affinity of the CD site by an order of magnitude, while the impairment of the CD site caused a dramatic drop in the effective Ca^2+^ affinity of the protein by five orders of magnitude.

Spectrofluorimetric titration of rWT α-PA by Sr^2+^ (Figure 2A) revealed quenching of Phe fluorescence down to the level observed upon addition of Ca^2+^ (Figure 1A), followed by a plateau at Sr^2+^ to a PA molar ratio above 1, and a reversal of the effect upon addition of EDTA. The description of the data by the single site metal binding scheme (3), considering the competition between the protein and EDTA for Sr^2+^ (Equation (2)) shows (Figure 2A and Table 1) that Sr^2+^ affinity of α-PA was two orders of magnitude lower compared to that for Ca^2+^. Meanwhile, the titration of rWT α-PA by Sr^2+^ monitored by ITC clearly demonstrates that the Sr^2+^-induced thermal effects extended up to Sr^2+^ and a PA molar ratio of 2 (Figure 2B). Approximation of the ITC data by the sequential metal binding scheme (1) gave two equilibrium Sr^2+^ association constants (Table 1), one of which was equivalent to the estimate from the fluorimetric titration, while the other was an order of magnitude lower. Hence, Phe fluorescence of rWT α-PA appears to be insensitive to binding of the second Sr^2+^ ion.

Impairment of CD/EF sites of α-PA was accompanied by Sr^2+^-induced fluorescence changes closely resembling those observed upon the Ca^2+^ titrations (Figure 3). While the damage of the EF site suppressed Sr^2+^ affinity of α-PA by an order of magnitude (Table 1), the impairment of the CD site lowered apparent Sr^2+^ affinity of the protein by 1.5 orders of magnitude and resulted in an inability to achieve the fluorescence intensity inherent to its Sr^2+^-loaded form. These results provide evidence that Sr^2+^ binds to CD/EF sites of rWT α-PA, confirming a pivotal role of the CD site for maintenance of Sr^2+^ affinity of the protein, and indicating the sequential mechanism of the Sr^2+^ binding, with the CD site being loaded first.

### 3.2. Secondary Sr^2+^-Binding Site(s) of Human α-Parvalbumin

Spectrofluorimetric titration of the Sr^2+^-loaded (100 μM Sr^2+^) rWT α-PA (12 μM) by Ca^2+^ up to 1 mM did not reveal noticeable effects (Figure 4A), and reflects nearly identical fluorescence parameters of the Sr^2+^/Ca^2+^-loaded proteins states. Meantime, spectrofluorimetric titration of the Ca^2+^-loaded (100 μM CaCl_2_) α-PA (1 μM) by Sr^2+^ up to a Sr^2+^ to Ca^2+^ molar ratio of 1:1 induced about a 10% increase in Phe fluorescence intensity (Figure 4B), which indicates the filling of a secondary Sr^2+^-binding site(s). An evaluation of the effective equilibrium Sr^2+^ association constant using single site metal binding scheme (3) gave ca. 4 × 10^5^ M^−1^.

### 3.3. Comparison of Structural Peculiarities of Sr^2+^/Ca^2+^-Loaded States of α-PA

Second-derivative FTIR spectroscopy spectrum of apo-α-PA exhibited some differences from those for the Sr^2+^/Ca^2+^-loaded protein at 20 °C (Figure 5). The bands centered around 1660 and 1550 cm^−1^ corresponded to Amide I and Amide II bands of peptide group, respectively, which overlap with contributions of protein side chains [46]. The FTIR spectroscopy second-derivative spectra of Ca^2+^- and Sr^2+^-loaded α-PA states were close to each other, suggesting similarity of their secondary and tertiary structures. The symmetric vibrations of protein COO^−^ groups located in the 1392–1425 cm^−1^ region [46] were similar for Ca^2+^/Sr^2+^-loaded α-PA forms, indicating similar states of their COO^−^ groups.

Loading of rWT α-PA with Sr^2+^/Ca^2+^ gave rise to near-equivalent decreases in molar ellipticity in the near-UV region (Figure 6A), indicating equal metal-induced increases in symmetry of microenvironments among the Phe residues. Similarly, Sr^2+^/Ca^2+^ association resulted in equal declines in molar ellipticity in the far-UV region (Figure 6B), indicating equivalent changes in the secondary structure of the protein. Analysis of the far-UV CD spectra using CDPro software [31] showed that Sr^2+^/Ca^2+^ binding increases α-helice content of α-PA from 30.6% up to 48–50% (Table 2). Overall, the CD results were in accord with the FTIR spectroscopy data (Figure 5).

Apo-α-PA exhibited a lack of a well-defined heat sorption peak within the temperature region from 20–120 °C, and its heat capacity approached that calculated for the fully unfolded protein (Figure 7), indicating the absence of a rigid tertiary structure, as previously demonstrated for pike α/β-PA (α/β isoform of parvalbumin) and coho salmon β-PA [20,26]. The heat sorption curve for Ca^2+^-loaded α-PA (1 mM CaCl_2_) reveals two superimposed heat sorption peaks at temperatures above 90 °C (Figure 7) similar to those found for Ca^2+^-saturated pike β-PA [33], which suggests the existence of two thermodynamic domains. In accordance with lowered Sr^2+^ affinity of α-PA relative to its affinity to Ca^2+^, the Sr^2+^-induced stabilization of the tertiary structure of the protein was less pronounced: Sr^2+^ binding gave rise to the appearance of a single heat sorption peak at 86 °C, which is well described within the cooperative two-state model (4) (Table 3).

### 3.4. Molecular Modeling

The fully optimized models of Ca^2+^/Sr^2+^-loaded CD and EF sites of PA are shown in Figure 8. Both metals were heptacoordinated, the glutamates in positions 62 (CD site) and 101 (EF site) bind Ca^2+^/Sr^2+^ in a bidentate mode, while the other carboxylic groups coordinated the metals in a monodentate manner, as observed in the X-ray structure 1PAL. The positive values of free energy changed upon substitution of Ca^2+^ for Sr^2+^ (ΔG^ε^ in Figure 8), confirming that Sr^2+^ binds more weakly to the both sites of PA than does Ca^2+^, in accord with the experimental observations (see Table 1). Notably, the free energies of metal exchange for the CD site were smaller than those for the EF site (1.2/1.6 versus 4.1/2.3 kcal/mol, respectively), which was in line with the experimental finding that the difference in Ca^2+^/Sr^2+^ affinities of the CD site (*K_a_* equals 4 × 10^9^ for Ca^2+^ and 2 × 10^7^ M^−1^ for Sr^2+^) is smaller than that for the EF site (*K_a_* equals 4 × 10^9^ for Ca^2+^ and 2 × 10^6^ M^−1^ for Sr^2+^). Furthermore, the theoretical results indicate that the metal competition for the CD site was slightly affected by changes in the dielectric constant (similar ΔG^4^ and ΔG^29^ values for buried and solvent-exposed binding sites, respectively; Figure 8A), whereas a decrease in the dielectric constant enhanced EF site protection against Sr^2+^ attack (ΔG^4^ exceeds ΔG^29^ in Figure 8B).

The abovementioned results were predicted for the fully optimized metal complexes, suggesting a certain flexibility of the CD/EF site to allow the incoming Sr^2+^ to affect the coordination geometry. How will the metal competition be affected in the case of a ‘rigid’ binding site? To answer this question, we substituted Ca^2+^ ions in the fully optimized sites with Sr^2+^, followed by single point calculations. The increase in the resultant free energies (shown in parentheses in Figure 8) provides evidence that the ‘rigid’ binding pocket favors its selectivity towards Ca^2+^. Note that EF-hand motifs are known to possess relatively stiff/inflexible structures that promote Ca^2+^ binding and increase selectivity towards Ca^2+^ [48].

Being bulkier compared to Ca^2+^, Sr^2+^ ions can accommodate more ligands in their first coordination sphere. To study the effect of an increased coordination number of Sr^2+^ on its competition with Ca^2+^ for the EF site of PA, we modeled a Sr^2+^ complex with this site where Sr^2+^ was 8-coordinated at the expense of an extra bond donated by either a bidentate aspartate (Figure 9A) or an additional water molecule (Figure 9B). The increase in the resultant free energies relative to the values for the heptacoordinated Sr^2+^ (Figure 8B) implies that in both cases the increase in coordination number of Sr^2+^ to 8 was unfavorable for competition with Ca^2+^.

The substitution of the bidenatate anionic Glu at position 12 of the EF-loop of PA for a monodentate neutral Gln decreased affinities of the binding site to Ca^2+^ and Sr^2+^ (Table 1). The data presented in Figure 10 suggest that despite the mutation-induced decrease in the metal affinity of the EF site, its metal selectivity did not suffer, since the ΔG values for PA_-EF_ (3.1/3.3 kcal/mol; Figure 10) overlap with the values for the wild-type PA (4.1/2.3 kcal/mol; Figure 8B).

## 4. Discussion

Rhyner et al. [21] studied Ca^2+^ binding to recombinant human α-PA and its mutants with inactivated CD and EF sites, but came to essentially different conclusions. While inactivation of the CD site did not affect Ca^2+^ affinity of the EF site, corruption of the EF site decreased Ca^2+^ affinity of the CD site. One of the factors, able to explain the discrepancy between our results and those obtained by Rhyner et al. could be use of different solvent conditions. For instance, it seems that elevated K^+^ level (150 mM KCl) was used by Rhyner et al. K^+^ ions are known to bind EF-hands of PA [49] and thereby interfere with other ionic equilibria. This suggestion can explain the Ca^2+^ association constants found by Rhyner et al. that were two orders of magnitude lower (2.3 × 10^7^ M^−1^).

The marked sensitivity of Sr^2+^ association with human α-PA to the amino acid substitutions inactivating its CD/EF sites (Figure 2 and Figure 3) gives evidence that the both EF-hands represent primary Sr^2+^-binding sites of the protein. Furthermore, the experiments with PA_-CD_ and PA_-EF_ mutants (Figure 1 and Figure 3) indicate that Sr^2+^/Ca^2+^ binding follows the same sequential mechanism of filling of the EF-hands, with the CD site being filled first. The sequential mechanism has been reported for other EF-hand proteins, including recoverin [50], Neuronal calcium sensor-1 [51], Calcium- and integrin-binding protein [52] and caldendrin [53]. Meanwhile, α-PA binds Sr^2+^ ions two to three orders of magnitude weaker than Ca^2+^ ions (Table 1). This result was intuitively expected, since the higher radius of Sr^2+^ compared to that for Ca^2+^ (1.18/1.21 Å versus 1.0/1.06 Å [10]) results in lower charge density of the metal and could geometrically prevent Sr^2+^ from filling the Ca^2+^-binding sites. The more sophisticated theoretical estimates fully confirm these predictions (Figure 8 and Figure 9), revealing that CD/EF sites of PA are well protected against the exchange of Ca^2+^ for Sr^2+^ regardless of coordination number of Sr^2+^, solvent exposure and rigidity of the sites. The lowered Sr^2+^ affinity compared to that for Ca^2+^ seems to be inherent to other EF-hand (calmodulin [54]) and non-EF-hand Ca^2+^-binding proteins, as exemplified by bovine α-lactalbumin [55].

Magnesium, another element from group II of Periodic table but positioned above calcium, has a much lower ionic radius compared to that for Ca^2+^: 0.72 Å versus 1.18 Å. Similarly to Sr^2+^, affinity of PAs for Mg^2+^ is several orders of magnitude lower than that for Ca^2+^ [19]. Thus, EF-hands of PA are selective towards calcium ions, which appears to be a consequence of inflexible structure of EF-hands that favors Ca^2+^ binding and their selectivity towards Ca^2+^ [48].

Contrary to Mg^2+^, which typically induces structural rearrangements resembling those observed upon Ca^2+^ binding but less pronounced (reviewed in [17]), Sr^2+^ binding to human rWT α-PA was accompanied by structural rearrangements unexpectedly close to those observed in response to Ca^2+^, as evidenced by the intensity of Phe fluorescence emission (Figure 1A and Figure 2A), FTIR spectroscopy data (Figure 5), near-UV molar ellipticity (Figure 6A), and far-UV CD data (Figure 6B, Table 2). A drastic difference between the Mg^2+^/Sr^2+^-induced effects could arise due to difference in coordination numbers of the metals: the smaller Mg^2+^ is typically characterized by six-coordinate geometry [34,56], while Sr^2+^ and Ca^2+^ are heptacoordinated (Figure 8). Nevertheless, Sr^2+^/Ca^2+^-loaded α-PA states differ from each other in stability of tertiary structure (Figure 7, Table 3). The Ca^2+^-loaded α-PA has two cooperative thermal transitions corresponding to the thermal unfolding of two thermodynamic domains, while the Sr^2+^-bound protein exhibits a single thermal transition corresponding to melting of a single cooperative domain. The difference in thermal behavior of Ca^2+^/Sr^2+^-loaded α-PA states likely reflects lowered Sr^2+^ affinity of the protein, which is insufficient for stabilization of the thermally induced intermediate state [33].

Despite similarities in coordination numbers of Ca^2+^ and Sr^2+^ and respectively similar structural consequences of their binding to α-PA, we have revealed a secondary site selective towards Sr^2+^, evidently different from primary Ca^2+^-binding sites of the protein (Figure 4B). Filling of the secondary site(s) with Sr^2+^ affects the microenvironment of the emitting Phe residues, inducing fluorescence intensity changes opposite to those observed upon filling of the primary sites with Sr^2+^/Ca^2+^. Since such secondary Sr^2+^-binding sites escape from competition with Ca^2+^, they represent the most direct way to interfere with the functioning of EF-hand proteins and, possibly, other proteins. Considering the multitude of functions performed by Ca^2+^-binding proteins, the structural alterations induced by filling of their Sr^2+^-selective sites may rationalize the variety of the side effects of strontium ranelate accompanying osteoporosis treatment (see Section 1).

## 5. Conclusions

The comparative experimental study of Sr^2+^/Ca^2+^ binding to recombinant human α-PA and its mutants with inactivated EF-loops have shown that both metals follow the same sequential mechanism of filling CD/EF sites, but Sr^2+^ affinity of the protein is 2–3 orders of magnitude lower relative to that of Ca^2+^. The latter effect is fully supported by theoretical predictions, which proved to be robust to level of solvent exposure and rigidity of the metal sites, and varying coordination numbers of Sr^2+^. Sr^2+^/Ca^2+^ binding to α-PA results in strikingly similar protein structures (likely owing to identical coordination numbers of the metals) that differ in stability of tertiary structure due to differences in Sr^2+^/Ca^2+^ affinities. The secondary Sr^2+^-selective site we discovered points to an opportunity for Sr^2+^ and its radioactive counterpart ^90^Sr^2+^ to affect functional activity of proteins without competition with Ca^2+^ for the same sites.

## Figures and Tables

**Figure 1 biomolecules-11-01158-f001:**
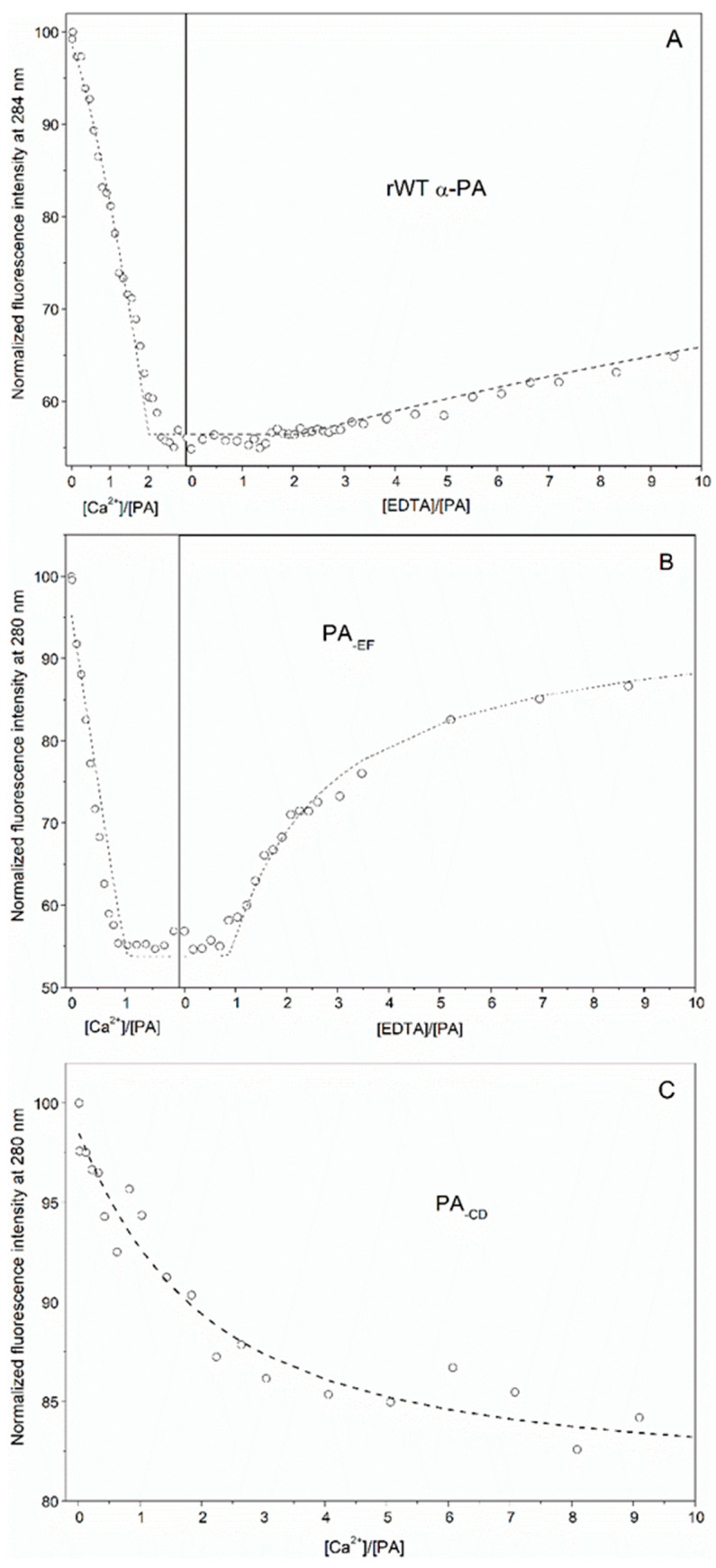
Spectrofluorimetric titrations of rWT α-PA (**A**) and its mutant forms, PA_-EF_ (**B**) and PA_-CD_ (**C**) via Ca^2+^/EDTA at 20 °C (10 mM HEPES-KOH, pH 8.1). Protein concentration is 12–17 μM. Fluorescence was excited at 259 nm. The points are experimental. (**A**) The curve is fit according to the sequential metal binding scheme (1), considering a competition between the protein and EDTA for Ca^2+^ (scheme (2)). (**B**,**C**) The curve is a fit within the single site metal binding scheme (3). The competition between the protein and EDTA for Ca^2+^ is taken into account for PA_-EF_.

**Figure 2 biomolecules-11-01158-f002:**
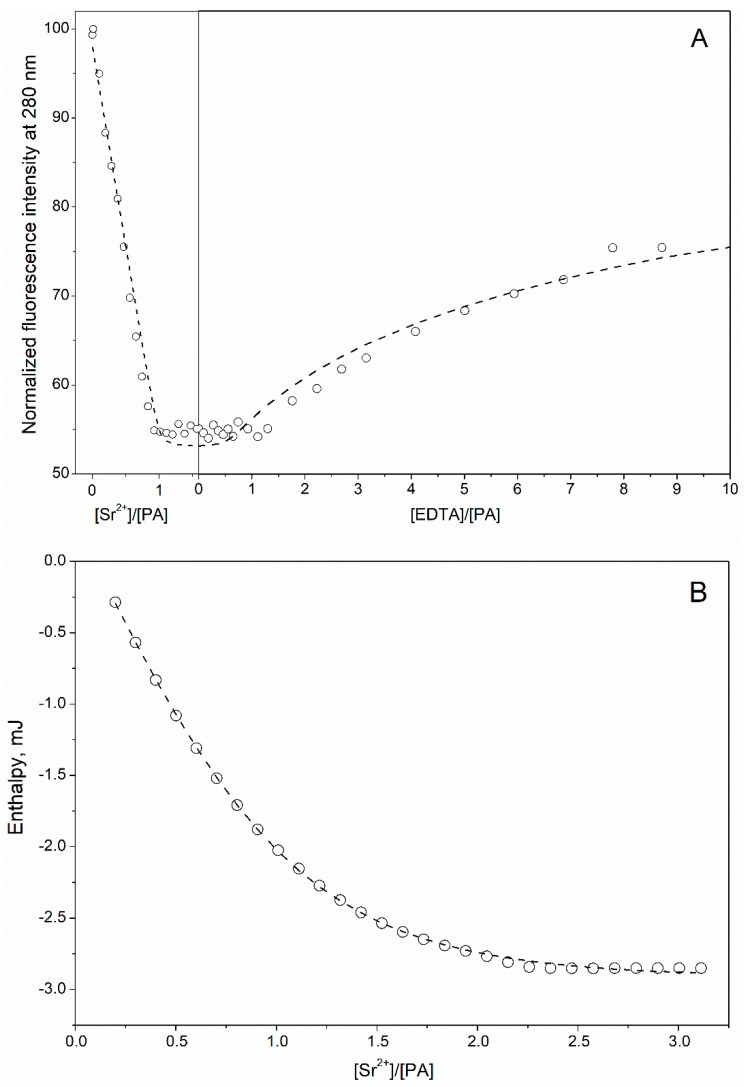
The interaction of Sr^2+^ with rWT α-PA at 20 °C, followed by Phe fluorescence (**A**) and ITC (**B**) (10 mM HEPES-KOH pH 8.1). The points are experimental. (**A**) Protein concentration is 19 μM. Fluorescence was excited at 259 nm. The data are approximated by the single site metal binding scheme (3), considering a competition between the protein and EDTA for Sr^2+^ (scheme (2)). (**B**) Protein concentration is 97 μM. The curve is a fit according to the sequential metal binding scheme (1).

**Figure 3 biomolecules-11-01158-f003:**
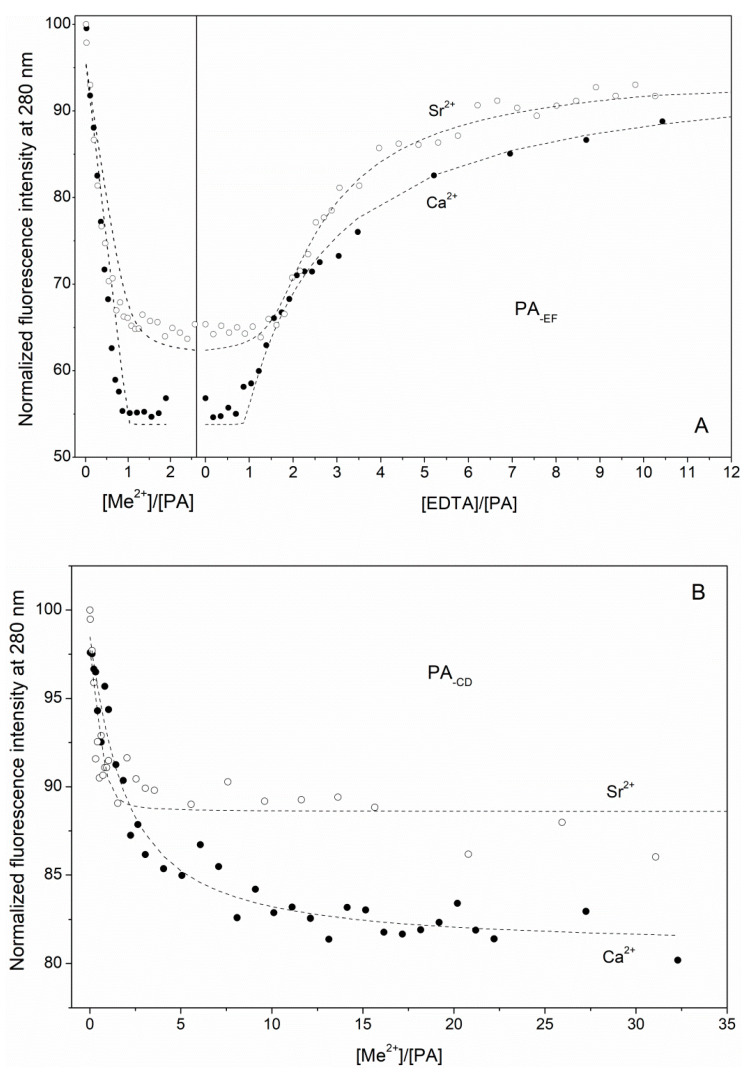
The interaction of Sr^2+^/Ca^2+^ with PA_-EF_ (**A**) and PA_-CD_ (**B**) at 20 °C, followed by Phe fluorescence (10 mM HEPES-KOH pH 8.1). Protein concentration is 14–19 μM. Fluorescence was excited at 259 nm. The points are experimental. The curves are approximated by the single site metal binding scheme (3). In the case of PA_-EF_, the competition between the protein and EDTA for metal ions (scheme (2)) is taken into account.

**Figure 4 biomolecules-11-01158-f004:**
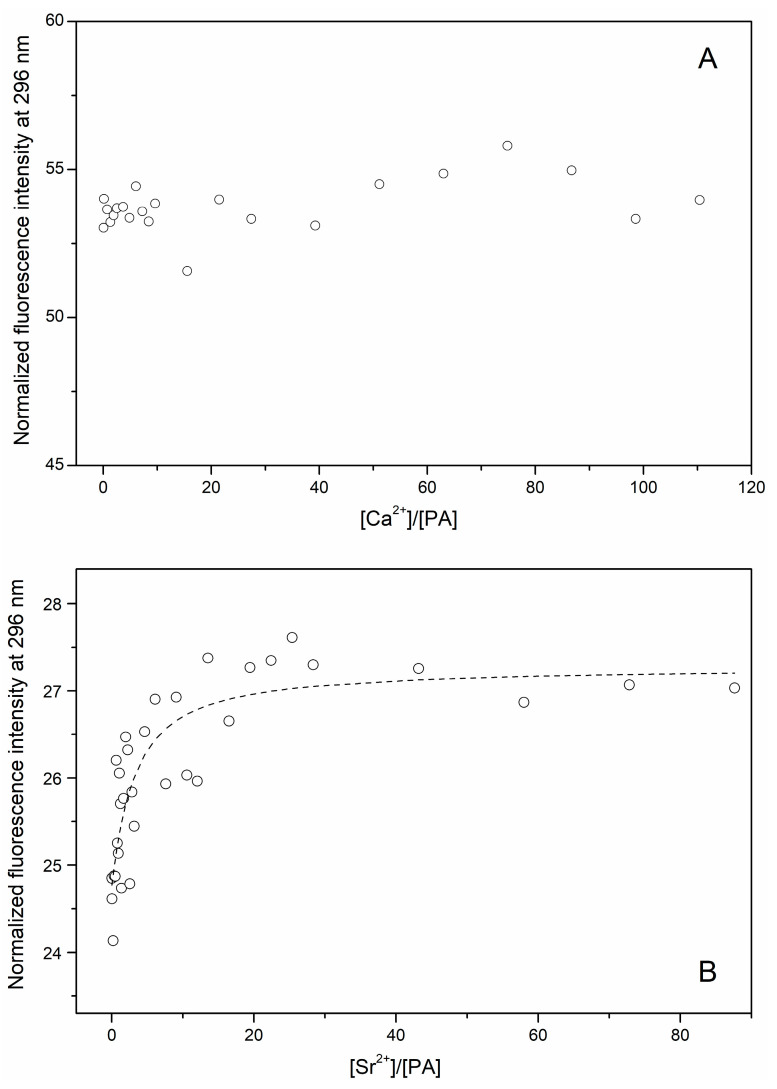
Spectrofluorimetric study of competition between Sr^2+^ and Ca^2+^ for rWT α-PA at 20 °C (10 mM HEPES-KOH, pH 8.1). Fluorescence was excited at 259 nm. (**A**) Titration of metal-depleted α-PA (12 μM) by Ca^2+^ in the presence of 100 μM SrCl_2_. (**B**) Titration of α-PA (1 μM) by Sr^2+^ in the presence of 100 μM CaCl_2_.

**Figure 5 biomolecules-11-01158-f005:**
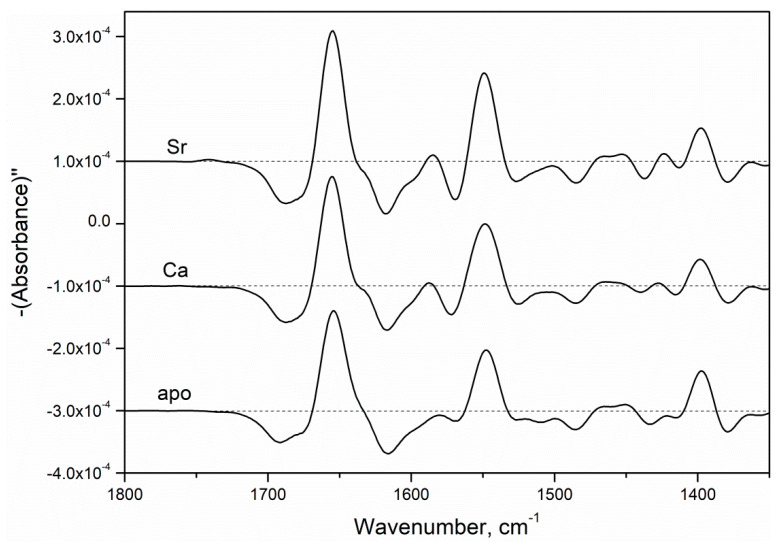
Second-derivative FTIR spectroscopy spectra of apo- and Sr^2+^/Ca^2+^-loaded (10 mM) α-PA (3 mM) at 20 °C (10 mM H_3_BO_3_-KOH, pH 9.0).

**Figure 6 biomolecules-11-01158-f006:**
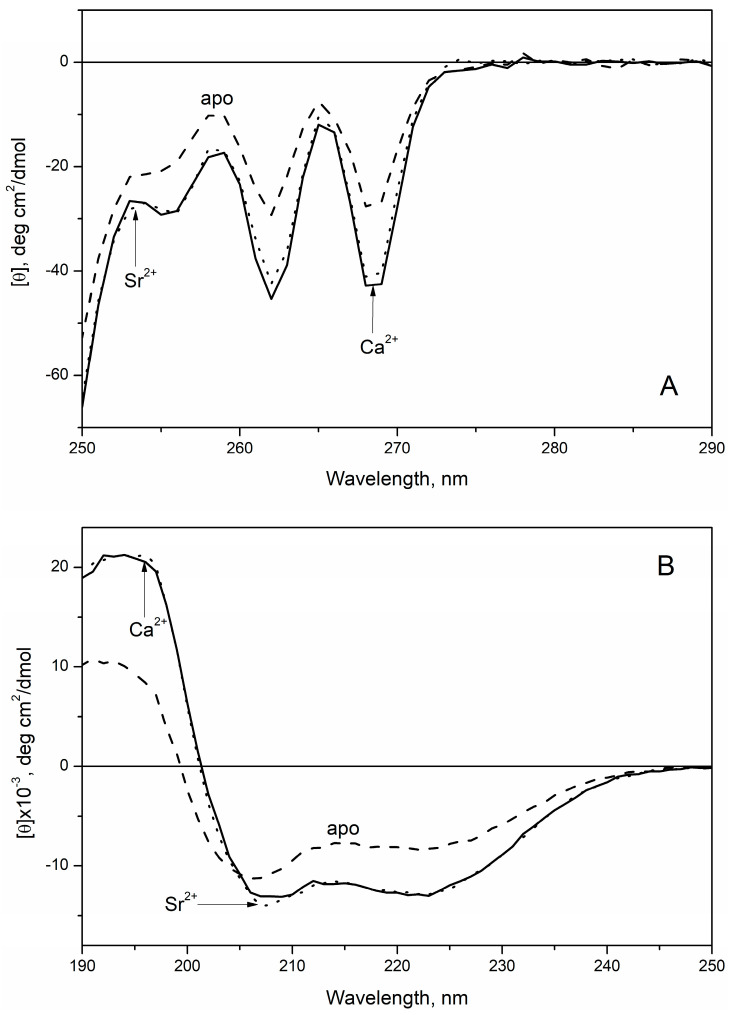
Near- (**A**) and far-UV (**B**) CD spectra for apo- and Sr^2+^/Ca^2+^-loaded states of rWT α-PA at 20 °C (10 mM H_3_BO_3_-KOH, pH 9.0, 1 mM EDTA or 1 mM SrCl_2_/CaCl_2_). Protein concentration was 6.2–7.5 and 126–129 µM for far- and near-UV regions, respectively.

**Figure 7 biomolecules-11-01158-f007:**
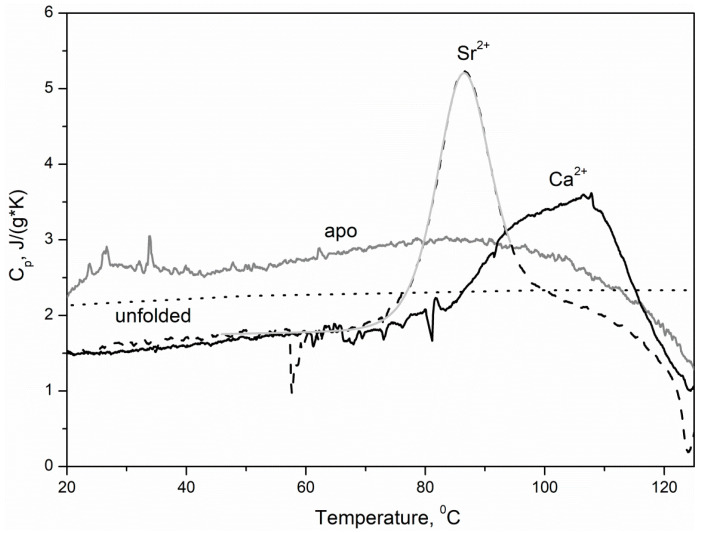
Temperature dependencies of specific heat capacities of metal-depleted and Sr^2+^/Ca^2+^-bound states of rWT α-PA, derived from DSC data according to [47]. Protein concentration is 1.2–1.8 mg/mL. Buffer conditions: 10 mM H_3_BO_3_-KOH pH 9.0, 1 mM SrCl_2_/CaCl_2_ or without additions. Heating rate is 0.86 K/min. The data for the Sr^2+^-loaded form were analyzed according to the cooperative two-stat*e* model (4). The theoretical curve is shown in light grey. The dotted curve corresponds to the specific heat capacity of the fully unfolded α-PA, calculated according to [32].

**Figure 8 biomolecules-11-01158-f008:**
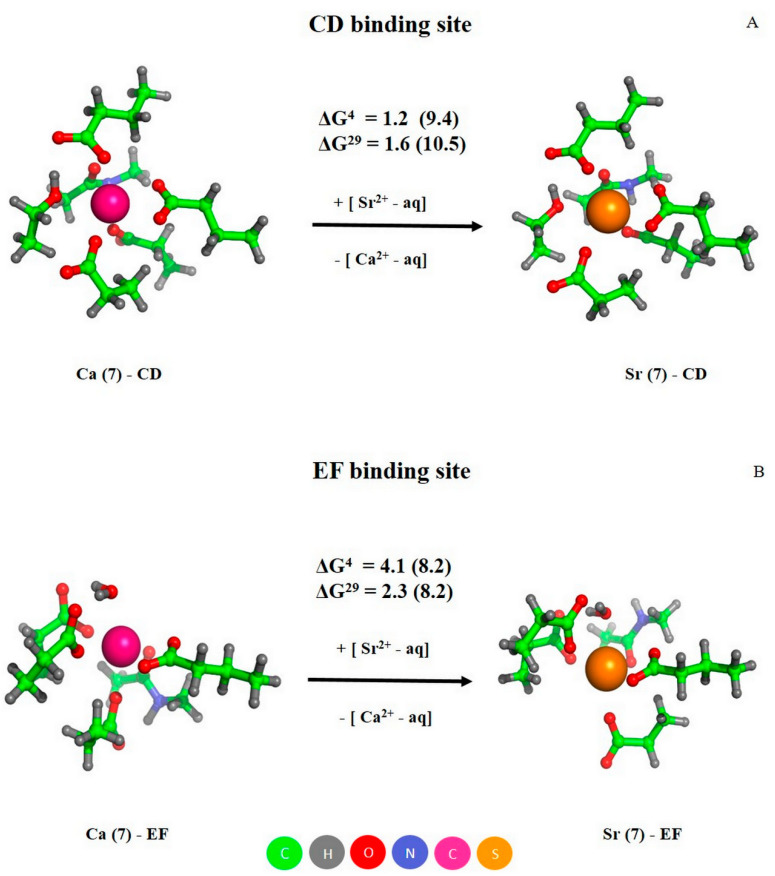
The optimized structures of Ca^2+^/Sr^2+^-loaded CD (**A**) and EF (**B**) sites of PA, and free energies of metal exchange (in kcal/mol). The free energies evaluated for ‘rigid’ Ca^2+^-binding sites are given in parentheses. Metal coordination number is indicated in parentheses in the complex name.

**Figure 9 biomolecules-11-01158-f009:**
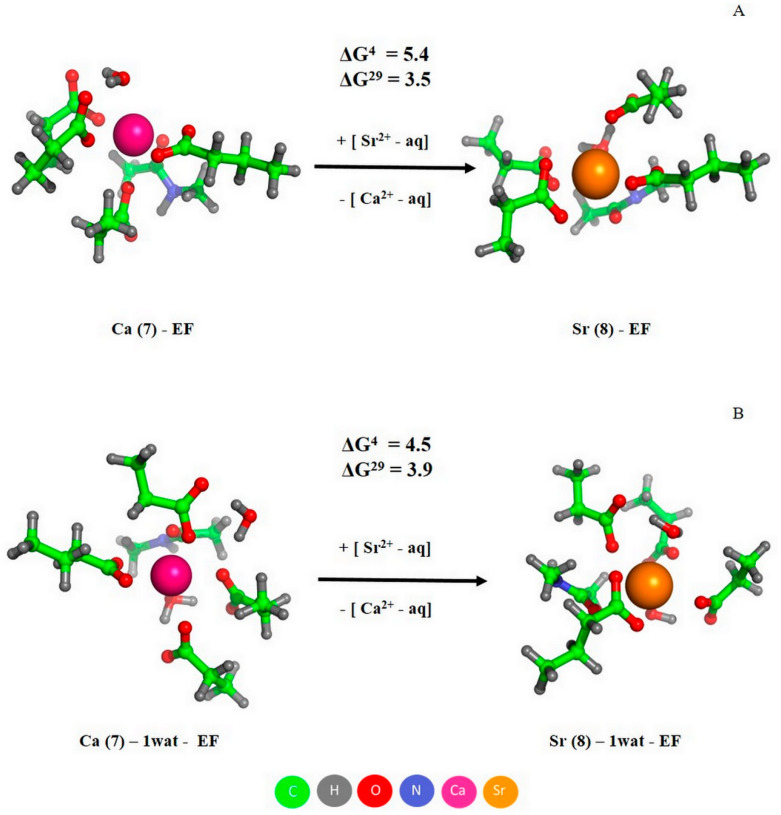
The optimized structures of Ca^2+^/Sr^2+^-loaded EF site of PA with 8-coordinated Sr^2+^, and free energies of metal exchange (in kcal/mol). Sr^2+^ acquires an additional bond at the expense of either bidentate Asp binding (**A**) or an additional water molecule (**B**). Metal coordination number is indicated in parentheses in the complex name.

**Figure 10 biomolecules-11-01158-f010:**
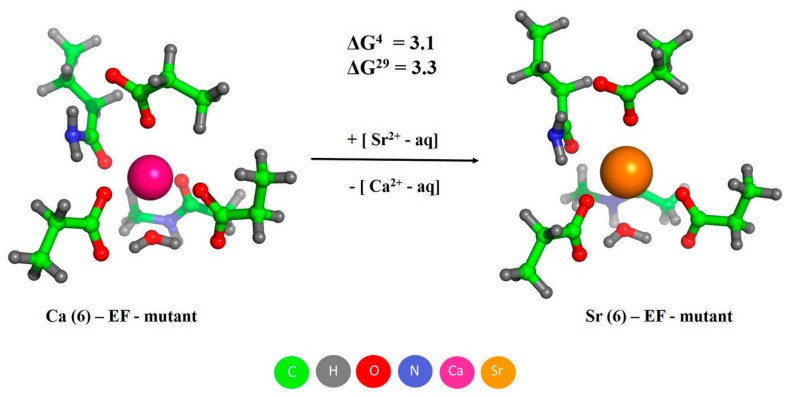
The optimized structures of Ca^2+^/Sr^2+^-loaded EF site of PA_-EF_, and free energies of metal exchange (in kcal/mol). Metal coordination number is indicated in parentheses in the complex name.

**Table 1 biomolecules-11-01158-t001:** The equilibrium Ca^2+^/Sr^2+^ association constants for rWT α-PA and its mutant forms at 20 °C, evaluated from the intrinsic fluorescence and ITC data, shown in Figure 1, Figure 2 and Figure 3 (10 mM HEPES-KOH, pH 8.1).

Ca^2+^		
Protein	Fluorimetric titrations	
	*K*_1_, M^−1^	*K*_2_, M^−1^
rWT	4 × 10^9^	4 × 10^9^
PA_-CD_	5 × 10^4^	
PA_-EF_	4 × 10^8^	
**Sr^2+^**		
**Protein**	**Fluorimetric titrations/ITC**	
	***K*** **_1_** **, M^−1^**	***K*** **_2_** **, M^−1^**
rWT	3 × 10^7^/(2 × 10^7^ and 2 × 10^6^)	
PA_-CD_	1 × 10^6^	
PA_-EF_	2 × 10^6^	

**Table 2 biomolecules-11-01158-t002:** The secondary structure fractions estimated for rWT α-PA at 20 °C from the CD data shown in Figure 6, using CDPro software [31].

Protein State	α-Helices, %	β-Sheets, %	Turns, %	Random Coil, %
apo	30.6 ± 0.2	17.6 ± 0.3	19.6 ± 0.3	31.50 ± 0.10
Sr^2+^	50.1 ± 0.2	7.4 ± 0.3	15.5 ± 0.3	26.80 ± 0.10
Ca^2+^	48.2 ± 0.2	8.6 ± 0.3	15.7 ± 0.3	27.30 ± 0.10

**Table 3 biomolecules-11-01158-t003:** Thermodynamic parameters of thermal denaturation of Sr^2+^-loaded α-PA estimated from the DSC data shown in Figure 6 according to the cooperative two-state model (4): mid-transition temperature (*T_0_*), enthalpy of protein denaturation at temperature *T_0_* (Δ*H_0_*), heat capacity change accompanying the transition (Δ*C_p_*) and number of molecules involved into transition (*n*).

T_0_, °C	ΔH_0_, J/g	ΔC_p_, J/(g·K)	n
86.0	37.5	0.40	0.81

## Data Availability

Not applicable.

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
