# Peer review of "Strontium Binding to α-Parvalbumin, a Canonical Calcium-Binding Protein of the “EF-Hand” Family"

_biomolecules, 2021, doi:10.3390/biom11081158_

Round 1

Reviewer 1 Report

The manuscript ” Strontium Binding to α-Parvalbumin, a Canonical Calcium-binding Protein of the “EE-hand” Family” by Vologzhannikova A.A. et al.  addresses an important problem concerning the mechanisms of strontium interactions with calcium-binding protein of EF-hand family such as parvalbumin. The authors conducted very careful studies of strontium binding to two binding sites in parvalbumin molecule and effect of critical residue Glu 12 substitution in the major binding site on the binding of both calcium and strontium. They have demonstrated that despite lowered affinity of parvalbumin to strontium, it still can compete with calcium for the same EF-hands and induces similar structural rearrangements. Strontium induces also increase in parvalbumin stability, though on smaller scale compared to the calcium-induced effect. Strontium salts are used for treatment of osteoporosis and bone cancer, therefore it is important to reveal their potential effect on calcium-mediated physiological processes and calcium-binding proteins, which the authors have clearly demonstrated. They have used a broad range of methods, including fluorometric and isothermal titration calorimetry, mutation analysis, circular dichroism in the near and far UV regions to assess the effect on protein tertiary and secondary structure, scanning calorimetric measurements and theoretical modeling using Density Functional Theory computations with Polarizable Continuum Model calculations. The results and conclusions are very convincing. The manuscript is clearly structured and well written. The figures are clear and of good quality. The manuscript is of interest for the scientific community and it is recommended for publication in ”Biomolecules” without reservations. 

On the positive side I would add that the technical performance and analysis of the results are to very high standards. That is consistently high quality data on metal-binding mechanisms and action from this laboratory.

Regarding some improvement of the manuscript, since strontium salts are used for treatment of osteoporosis and bone cancer, it would be beneficial to add a paragraph to the discussion regarding which implications the new findings of strontium binding to calcium-binding protein parvalbumin may have on the usage of strontium in therapeutic treatments and understanding of its mechanisms of therapeutic effects.

Author Response

Response: We have added the following text to the very end of the Discussion chapter: “Considering the multitude of functions, performed by Ca2+-binding proteins, the structural alterations induced by filling of their Sr2+-selective sites may rationalize the variety of the side effects of strontium ranelate accompanying osteoporosis treatment (see Introduction chapter)”.

Reviewer 2 Report

This work reports the strontium binding mechanism for an EE-Hand Family model protein, compared to calcium. Using a large set of biophysical techniques, authors describe Sr2+ binding stoichiometry, affinity and secondary binding sites. Finally, authors propose the structural rigidity of the binding sites as the cation selectivity mechanism for this protein.

Although a large amount of work was performed, the reviewer would be glad to recommend this paper for publication once the following minor revisions are done.

  1. In the title, replace “EE-hand” by “EF-hand”.
  2. When authors monitors the Ca2+ binding process by Phe fluorescence and uses EDTA to invert the effect, the transition in the case of rWT–αPA seems not fully reversible to me (Figure 1A). Could this be due to the presence of protein oligomers, aggregates or even fibers in the sample upon depletion?. Are the authors considering the Thioflavin T fluorescence assay to identify, if any, the presence of amyloid fibrils in samples, as has been observed by other authors in ß-PA (Martínez et al. 2015. Swiss Med Wkly. 2015;145:w14128)?. Independently of this, in Figure 1A the fluorescence intensity encoded in the graph is 284 nm while in B and C panels is 280 nm, why?.
  3. In Figure 6B, the Sr2+ curve (light grey) is hard to see. Consider an alternative and clearer representation.
  4. Authors start the discussion citing the work of Rhyner et al in which different conclusions regarding the sequential occupancy of CD and EF sites is proposed. For the authors, the possible explanation for this discrepancy is the presence of 150 mM KCl in Rhyner’s experiments, and the interference between cations in the binding equilibria. As αPA is an intracellular protein, and 150 mM KCl represents a physiological concentration for K+, it will be desirable to perform a titulation in presence of the two ions (Ca2+/K+), same as have been done with Sr2+, using Phe fluorescence on the wt, EF and CD mutant.
  5. Finally, in the discussion, it will be interesting if authors could speculate/develop a little bit more about the adverse effects of treatments such as aforementioned strontium ranelate and its relation with Sr2+ EF-Hand family protein binding. Martínez et al. 2015. Swiss Med Wkly. 2015;145:w14128 demonstrates the correlation between amyloid formation in ß-PA and IgE allergic response. The saturation of the Ca2+ binding sites in those proteins is crucial for the IgE epitope recognition (Kuehn et al. 2014. Front. Immunol. https://doi.org/10.3389/fimmu.2014.00179). Maybe the saturation of alternative binding sites with Sr2+ could increase this allergic response in patients?.

Author Response

Response 1: done.

Response 2: In fact, the titration curve shown in Figure 1A is fully reversible. To visualize stoichiometry of metal binding to PA, we have shown initial part of the curve, but removed the rest of the curve (up to EDTA to protein molar ratio of 130).

Since the protein solutions were fully transparent regardless of the metal content, we had no reasons to suspect formation of protein aggregates and/or fibrils.

Response 3: The both wavelength are equally close to Phe fluorescence spectrum maximum position. Therefore, they are equally suited for the analysis.

Response 4: done.

Response 5: In fact, the work by Rhyner et al does not specify solution conditions used in the metal binding experiments, so the discrepancy between our results and those obtained by Rhyner et al could be due to other factors, including choice of buffer and use of other solutions additives.

Addition of 150 mM KCl in our titration experiments greatly complicates analysis of the data due to necessity to take into consideration competition of potassium ions for Ca2+/Sr2+-binding sites of the protein. We intentionally avoided such complications to facilitate analysis and interpretation of the data.

Response 6: We have added the following text to the very end of the Discussion chapter: “Considering the multitude of functions, performed by Ca2+-binding proteins, the structural alterations induced by filling of their Sr2+-selective sites may rationalize the variety of the side effects of strontium ranelate accompanying osteoporosis treatment (see Introduction chapter)”.